# Sub-nanometer confinement enables facile condensation of gas electrolyte for low-temperature batteries

Guorui Cai[1,6], Yijie Yin[2,6], Dawei Xia[3,6], Amanda A. Chen [1,3], John Holoubek[1], Jonathan Scharf[1], Yangyuchen Yang[2], Ki Hwan Koh[1], Mingqian Li[3], Daniel M. Davies[1], Matthew Mayer[1], Tae Hee Han [4], Ying Shirley Meng[1,2,5], Tod A. Pascal [1,2,3,5] & Zheng Chen [1,2,3,5 ✉]

Confining molecules in the nanoscale environment can lead to dramatic changes of their physical and chemical properties, which opens possibilities for new applications. There is a growing interest in liquefied gas electrolytes for electrochemical devices operating at low temperatures due to their low melting point. However, their high vapor pressure still poses potential safety concerns for practical usages. Herein, we report facile capillary condensation of gas electrolyte by strong confinement in sub-nanometer pores of metal-organic framework (MOF). By designing MOF-polymer membranes (MPMs) that present dense and continuous micropore (~0.8 nm) networks, we show significant uptake of hydrofluorocarbon molecules in MOF pores at pressure lower than the bulk counterpart. This unique property enables lithium/fluorinated graphite batteries with MPM-based electrolytes to deliver a significantly higher capacity than those with commercial separator membranes (~500 mAh g$^{-1}$ vs. <0.03 mAh g$^{-1}$) at −40 °C under reduced pressure of the electrolyte.

[1] Department of NanoEngineering, University of California, San Diego, CA, USA. [2] Program of Materials Science and Engineering, University of California, San Diego, CA, USA. [3] Program of Chemical Engineering, University of California, San Diego, CA, USA. [4] Department of Organic and Nano Engineering, Hanyang University, Seoul, Republic of Korea. [5] Sustainable Power and Energy Center, University of California, San Diego, CA, USA. [6] These authors contributed equally: Guorui Cai, Yijie Yin, Dawei Xia. ✉email: zhengchen@eng.ucsd.edu

**B**atteries that can sustain ultralow temperatures (<−30 °C) are essential for extending the operation capability of existing energy storage systems as well as enabling human presence to the outer space and deep ocean worlds[1–5]. The state-of-the-art lithium-ion batteries (LIBs) are mostly restricted to perform in mild conditions due to the drastically decreased ionic conductivity and increased charge-transfer impedance of electrode/electrolyte interfaces at ultralow temperatures[1–11], despite that some cells like lithium-thionyl chloride batteries are capable of operation down to −80 °C for low power applications[12–14]. Although many approaches (e.g., externally/internally heating, cell insulating, and introducing co-solvents or additives) have been developed to overcome the above issues[5–11], no current technology extends the operating temperature range of batteries without sacrificing the long-term stability and energy density.

Unlike conventional liquid/solid electrolyte chemistries, liquefied hydrofluorocarbon gas molecules like fluoromethane (FM) show a low melting point (−142 °C) and low viscosity (0.085 mPa·s), which enable electrolytes with high ionic conductivity and superior lithium metal compatibility down to temperatures as low as −60 °C[15–17]. To retain the electrolyte in the liquid state, the gas molecules need to be maintained at their vapor pressure ($P_v$). However, the saturated vapor pressure for these gas molecules is very high ($P_{sat, FM} = 495$ psi or 33 atm at 20 °C), which would render safety concerns in practical devices.

To address the above-mentioned limitation, it is conceivable to exploit capillary condensation, a phenomenon whereby gas molecules in small confined pores condense into a liquid at an equilibrium pressure $P_v$ that is lower than the bulk vapor pressure $P_{sat}$[18,19]. The relationship between the $P_v$ and the $P_{sat}$ follows the Kelvin equation:

$$\ln \frac{P_v}{P_{sat}} = \frac{2\gamma V_L}{rRT} \tag{1}$$

where $\gamma$ is the liquid/vapor surface tension, $V_L$ the molar volume of the liquid adsorbate, $r$ the mean radius of curvature of the liquid/gas interface (proportional to pore radius), $R$ the universal gas constant, and $T$ the absolute isothermal temperature. Generally, smaller $r$ values enable a lower $P_v$ to condense gas molecules at a given temperature (Fig. 1)[20–22]. For example, both

simulation and experimental results show that the actual pressure required to condense nitrogen molecules in porous carbon (slit-shaped pores) reduces by >10 times as the pore diameter decreases from 7 to 1 nm[23]. Similar trends have also been observed for methane absorption and condensation in nanopores[24]. Hydrofluorocarbons share some similar physicochemical characteristics with methane (e.g., high vapor pressure, low melting point), yet capillary condensation of hydrofluorocarbon has neither been explored, nor has there been a study reporting the design of a stable nanoporous host to condense gas molecules at reduced pressures for electrochemical applications.

Metal-organic frameworks (MOFs), a class of porous crystalline solids assembled by organic linkers and metal ions/clusters, could be ideal candidates to capture FM molecules via capillary condensation to lower the inner pressure of batteries employing liquefied gas electrolytes (LGE)[25–27]. Various MOFs have been successfully applied for the storage or separation of carbon dioxide, methane, and alkene molecules[24,28–30]. Composites formed from MOF particles and polymer binders have been employed as separator membranes of LIBs for an increased Li+ transference number based on the size-selective effect of MOF pores[31–37]. However, such MOF-based membranes are not suitable for desired ion migration in LGE under reduced pressure, due to the presence of numerous gaps between the binders and MOF particles, which inevitably degrade the continuous liquefied gas flux required for Li+ migration throughout the entire membrane.

Pure MOF membranes have been reported for gas adsorption/separation and solid electrolyte[28–31]; however, their poor mechanical properties and large thickness (typically > 100 μm) inhibit their applications in batteries. MOF particles have therefore been combined with commercial Celgard membranes or other porous substrates to improve the mechanical properties of the resulted MOF-based membranes[33,37]. However, additional substrates increase the separator thickness and hinder the cathode–anode ion pathway at reduced pressures. Therefore, it is prerequisite to develop a mechanically robust, self-supporting MOF-based porous membrane with small thickness and minimal macro-voids for continuous ionic transport in LGE, especially at reduced pressures.

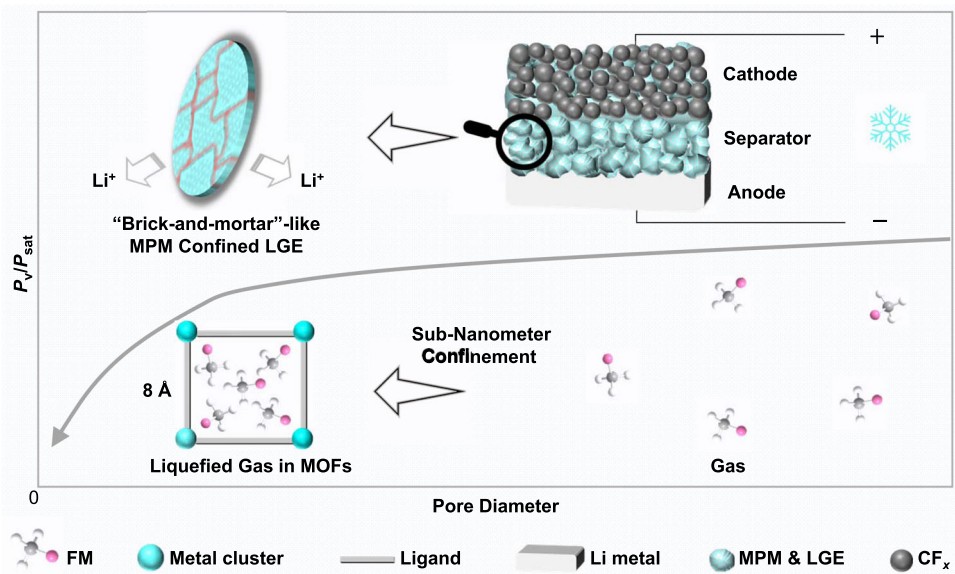

**Fig. 1 Schematic showing the mechanism of nano-confinement effects for lowering the equilibrium pressure of liquefied gas and the implementation of MPM-based LGE for Li-battery cells.** The cells were assembled by using LGE soaked MPM as the separator membrane, in which MOFs were employed as the porous hosts to condense the gas molecules under a lower $P_v$ than $P_{sat}$, attributed to the driving force of the sub-nanometer confinement of MOFs[61].

Herein, we demonstrated capillary condensation of FM gaseous electrolytes by designing a class of flexible MOF-polymer membranes (MPMs), which offered dense and continuous microporous (0.8 nm) networks (Fig. 1). The nano-confinement effects of MPMs toward FM molecules were systematically investigated by experiments and computer simulations. Leveraging the remarkable capillary condensation from the sub-nanometer pores in MPMs, we showed that LGE was able to operate below its vapor pressure. Decent ionic conductivity was achieved by using MPMs, which was impossible for conventional Celgard membranes. The corresponding $Li//CF_x$ (fluorinated graphite) cells employing such MPM-based gas electrolytes delivered a significantly higher capacity than those with commercial Celgard membranes (~500 mAh $g^{-1}$ vs. <0.03 mAh $g^{-1}$) at −40 °C and 70 psi. This study not only provides the insights on the molecular behavior in the nano-confined environment but also opens a potential pathway for safer operation of gas electrolytes.

## Results

**Rational selection of MOFs.** In order to identify a suitable MOF as the building blocks of MPM, a series of MOFs (HKUST-1, MOF-808, UiO-67, and UiO-66) with large pore volumes were tested and compared in terms of chemical/structural stability with FM[38–40]. To determine compatibility, as-synthesized MOF powders were subject to various characterizations before and after soaking in liquified FM. From $N_2$ sorption isotherms and the associated X-ray diffraction (XRD) patterns, it was found that all the selected MOFs maintained their crystallinity and highly microstructure features after exposure to FM, even at high pressure (~500 psi, room temperature) (Supplementary Figs. 1–4).

To further probe their potential as battery separators, a common solution-casting approach of mixing MOF powders and poly(vinylidene fluoride-co-hexafluoropropylene) (PVDF-HFP) binder solutions was used to produce MOF-based mixed matrix membranes (MMMs) (Supplementary Figs. 5–7). Note that from the fabrication point of view, such a method provides a simple and consistent way to screen different MOFs. The ionic conductivities in liquefied FM-based electrolytes were tested by using each corresponding MMM as a separator membrane (Supplementary Fig. 8). It was revealed that among various MOFs, UiO-66, and UiO-67 based MMMs provided the highest ion conductivity. Of note, at −60 °C the UiO-66 MMM exhibited an ionic conductivity of 0.67 mS $cm^{-1}$ while UiO-67 exhibited 0.75 mS $cm^{-1}$, higher than that of Celgard 2400 (0.36 mS $cm^{-1}$) with LGE. When the temperature was increased to 20 °C, UiO-66 maintained a conductivity of 0.54 mS $cm^{-1}$, much higher than that of Celgard (0.16 mS $cm^{-1}$) and UiO-67 (0.35 mS $cm^{-1}$). Based on the confirmed chemical stability in the presence of liquid FM and high ionic conductivity with LGE, UiO-66 was selected as the host structure for subsequent development of MPM.

**Synthesis and characterization of UiO-66-based MPMs.** Despite the high ionic conductivity at high operating pressure, it was observed that large stacking pores (micron scale) between MOF particles persisted in MMMs (Supplementary Fig. 7), which would prohibit a continuous liquefied gas flux for the $Li^+$ migration throughout the whole membrane beneath the vapor pressure, due to few solvents outside MOF pores to cover the gaps. In addition, MMMs showed appreciable mechanical degradation in liquefied FM due to the loosely connected MOF particles. To address the above issues, a dense and flexible microporous membrane was further designed. Inspired by the natural process in mollusks to fabricate bulk synthetic nacre[41,42],

"brick-and-mortar"-like MPMs were successfully fabricated by using 2-dimensional (2D) graphene oxide (GO)@UiO-66 nanosheets as the porous "brick".

Typically, driven by the abundant oxygen functional groups (e.g., –OH and –COOH) on GO, UiO-66 particles were in situ grown on GO (as a structure-directing template) in dimethylformamide (DMF) to achieve 2D GO@UiO-66 nanosheets. After mixing with PVDF (20 wt%), a free-standing MPM was obtained through a solution-casting process (Supplementary Fig. 5) followed by rolling to compact the MPM (Fig. 2a). As shown in Fig. 2b, the GO@UiO-66 exhibited a Brunauer-Emmett-Teller (BET) surface area of 654 $m^2$ $g^{-1}$. After integrating GO@UiO-66 with a PVDF binder, the resulting free-standing MPM maintained the high BET surface area (436 $m^2$ $g^{-1}$) (Fig. 2b), indicating negligible blocking effect between the MOF pores and the polymer binder. The pore size distribution profiles showed a dominant pore diameter of the MPM at 0.8 nm (Fig. 2c), which was consistent with standard crystal structure of UiO-66. Powder XRD patterns revealed that the crystallinity of UiO-66 in the resulting MPM was also maintained during the solution-casting process (Fig. 2d). The scanning electron microscopic (SEM) images manifested that the UiO-66 particles were uniformly distributed on the GO surface and the GO@UiO-66 preserved the 2D structure of GO template (Fig. 2e). The top-view SEM image presented a smooth and even surface structure of a representative MPM (Fig. 2f). The cross-sectional SEM image exhibited a dense, crack-free structure with a uniform thickness (Fig. 2g and h).

To further explore structural information from the interior of MPM, three-dimension (3D) structure of a representative sample was reconstructed by X-ray nano-computed tomography (nano-CT, Fig. 2i)[43]. The 3D images demonstrated a homogeneous and continuous percolation network, which indicated their promise to allow $Li^+$ diffusion below the vapor pressure of FM. The dynamic reconstruction of the MPM via nano-CT can be found in the supporting videos (Supplementary Movie 1 and Movie 2), further confirming the continuous, dense and homogeneous interior structures of MPM. It is worth mentioning that the thickness of the resulted MPM can be easily adjusted by tuning the solution-cast process. For example, an MPM film with thickness of <30 μm (Fig. 2j) can be readily obtained using a lab-scale blade coater.

Additionally, the MPM presented good mechanical flexibility and impact resistance (Supplementary Fig. 9), attributed to the unique structural strengths of "brick-and-mortar"-like 2D architecture with lean polymer binder. To further check the stability of MOF-based membranes in FM, both UiO-66-based MMM and MPM were soaked in liquefied FM electrolyte for three days. Their morphologies after soaking were determined via SEM, in which the MMM exhibited an obvious collapse with large exposed pores (Supplementary Fig. 10). On the contrary, the MPMs retained a compact and even surface morphology (Supplementary Fig. 11), which further highlighted the significantly improved structural strength of the "brick-and-mortar" structure based on 2D MOF-based nanosheets compared with random network of MOF particles. All the above results proved that the self-supporting MPMs exhibited robust structure with high microporosity, which were highly desired as an LGE host.

**Microscopic interactions between FM and UiO-66.** The $N_2$ sorption isotherms of UiO-66 after soaking in liquid FM (Supplementary Fig. 4c) exhibited an increased surface area and pore volume, indicating an activation process of FM due to the strong interaction between FM and the open metal sites in UiO-66, similar to the supercritical $CO_2$ activation of MOF pores[30]. From the Fourier transform-infrared (FT-IR) spectrum (Fig. 3a), it was

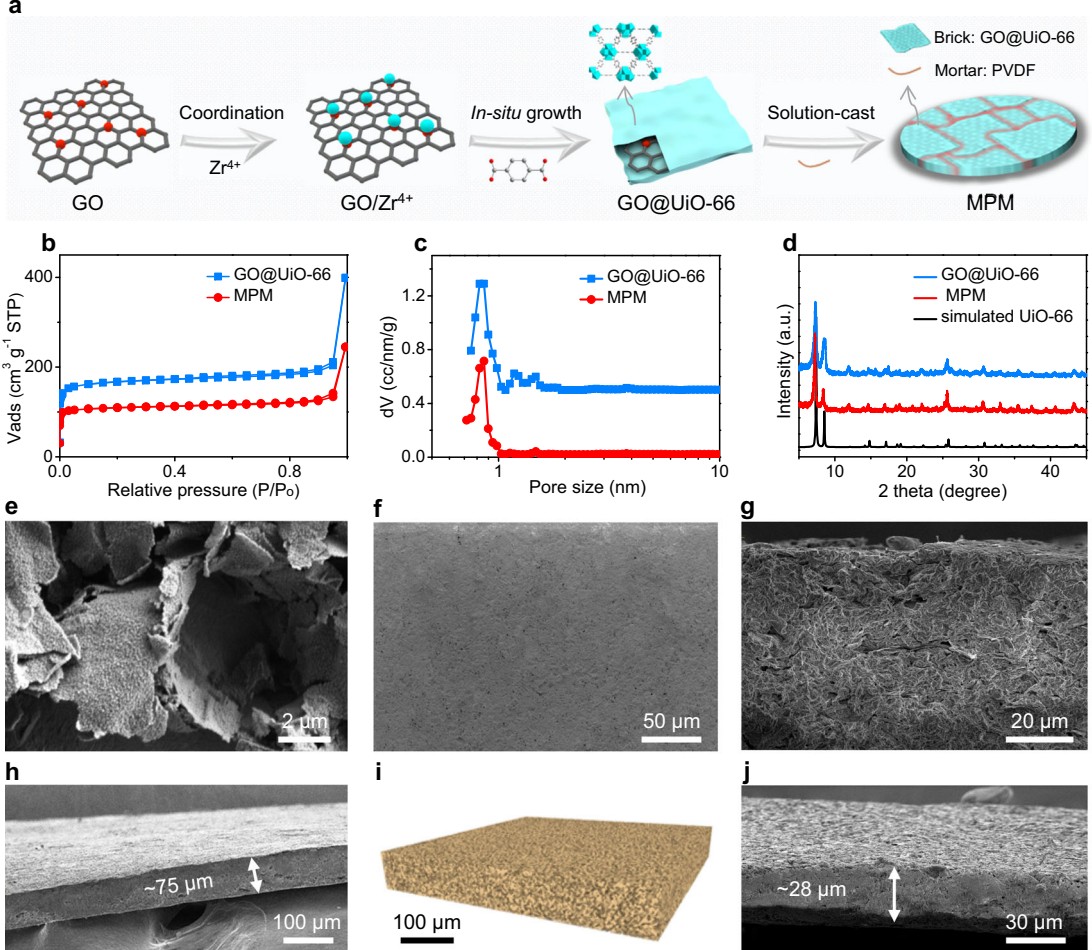

**Fig. 2 Characterization of MPMs. a** Illustration of fabrication process of MPMs, in which GO@UiO-66 nanosheets and PVDF binder working as the brick and mortar respectively. N$_2$ sorption isotherm curves (**b**), pore size distributions (**c**), and XRD patterns (**d**) of simulated UiO-66, GO@UiO-66, and MPM fabricated based on GO@UiO-66. **e** SEM images of GO@UiO-66. Top (**f**) and cross-sectional (**g**, **h**) SEM views of a representative MPM with different magnifications. **i** Nano-CT reconstruction of MPM. **j** Cross-sectional SEM view of a representative thin MPM.

observed that the symmetric COO⁻ ($\nu(COO^-)_{sym}$) stretching mode at around $1392\,cm^{-1}$ broadened after FM soaking (full width at half maximum (FWHM): 47.8 vs. 34.3), which indicated strong chemical interactions between the UiO-66 and FM[44,45]. This was further supported by the Raman spectra, in which the peak of the symmetric stretching mode at around 1444 and $1428\,cm^{-1}$ exhibited a change of shape (Fig. 3b)[44]. In addition, the signal of DMF molecules presented in theUiO-66 gradually decreased with the increase of soaking processes, indicative of the replacement of DMF with FM (Fig. 3a). It was also observed that the Raman spectra of FM-soaked samples exhibited a characteristic peak at $1000\,cm^{-1}$, attributed to the C-F vibration of the FM (Fig. 3b and Supplementary Fig. 12), indicating FM was still present in the UiO-66 pores even at ambient pressure. All the above results revealed that UiO-66 was able to firmly confine FM molecules beneath their vapor pressure.

**The adsorption behavior of FM in UiO-66.** In order to quantify the FM absorption capability of UiO-66, a direct measurement of FM uptake in the UiO-66 powders was performed after immersing the UiO-66 powders into FM with various pressures for three days. After purging out the bulk FM from the customized high-pressure cell, the total mass of the soaked UiO-66 powders was in situ measured in an Ar-filled glove box under ambient pressure. The mass difference of UiO-66 before and after the soaking

process was considered as the mass change caused by the absorbed FM. As shown in Fig. 3c, the mass of the UiO-66 powders increased by ~12% after soaking at ~500 psi, demonstrating the ability of UiO-66 to store of a large volume of liquefied FM molecules (corresponding to molar ratio of FM: UiO-66 at 5.7:1 for the absorbed sample). It is worth noting that the liquid/vapor surface tension can be further tuned by chemical modification of the MOF skeleton. As presented in Supplementary Fig. 13, UiO-66-NO$_2$ (a UiO-66 analogue with additional –NO$_2$ functional group) with a polar group on the MOF skeletons exhibits a little bit higher uptake capacity to confine the FM gas and slower release rate, while UiO-67 (a UiO-66 analogue with extended linkers), with increased pore sizes compared with UiO-66, poses reduced retention times due to a weak nanoconfinement effect. Considering the high complexity for simulating a variety of pore structures and chemical moieties, we select UiO-66 as the model system.

Further insights into the microscopic interactions between FM and UiO-66 were acquired from computer simulations (Fig. 3d). Both quantum mechanics (QM) calculations and molecular dynamic simulations were applied. In Supplementary Table 1, we described the intermolecular and intramolecular parameters of UiO-66 and FM, where the FM properties were obtained via QM calculations at the MP2/aug-cc-pVTZ level of theory using the Q-Chem 5.0 electronic structure package[46]. Initially, we optimized

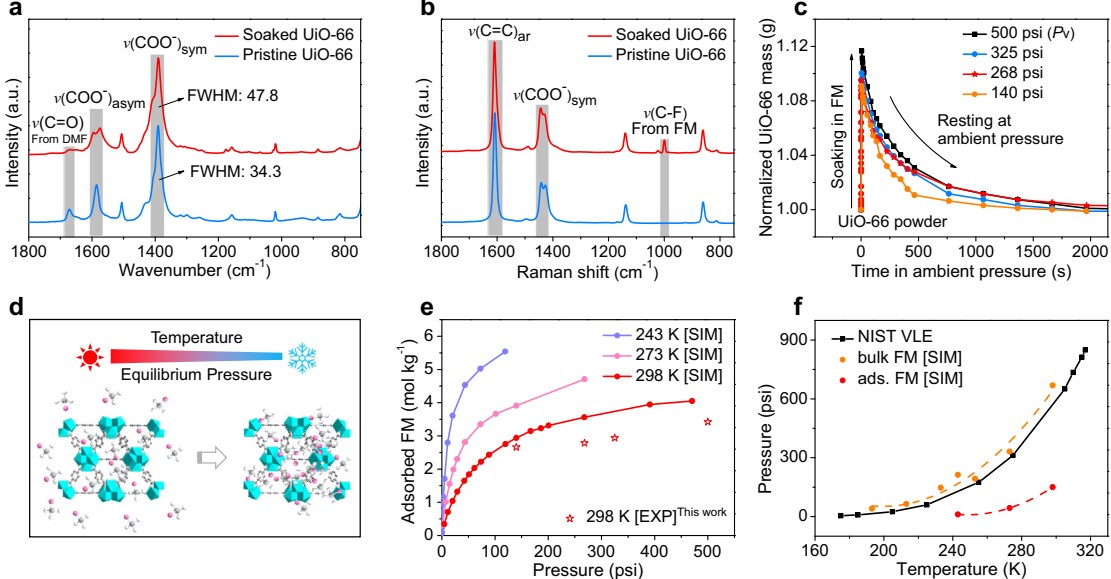

**Fig. 3 Interaction and adsorption behavior of FM in UiO-66.** FT-IR (**a**) and Raman spectrum (**b**) of UiO-66 before and after soaking in the liquefied FM. **c** Mass change of liquefied FM-soaked UiO-66. **d** Schematic showing the adsorption behavior of FM in UiO-66 at different conditions. **e** Simulated (SIM) adsorption isotherms of UiO-66 confined FM at different temperatures and experimental (EXP) results at 25 °C. **f** Phase transition comparison between bulk FM taken from NIST database, simulated (SIM) bulk FM and simulated (SIM) adsorbed (ads.) FM in UiO-66 systems, where the orange and the red points represent as the phase transitions of bulk FM and adsorbed FM (ads. FM) in UiO-66, respectively.

the UiO-66 starting structure (Supplementary Fig. 14) using molecular dynamics (MD) simulations via the Large-scale Atomic/Molecular Massively Parallel Simulator (LAMMPS) simulation engine[47]. The loading curves of FM in UiO-66 were then obtained from the optimized structure by means of Grand Canonical Monte-Carlo (GCMC) simulations using the MCCCS Towhee simulation package[48]. The accuracy of our GCMC approach was confirmed by comparing the adsorption isotherms of $CH_4$ and $CO_2$ to other published works as shown in Supplementary Figure 15[49,50]. All simulated adsorption isotherms of FM in UiO-66 at variable temperatures exhibited a classical type I isotherm of micropore adsorption (Fig. 3e), in which UiO-66 achieved a 10% mass uptake of FM at 140 psi and 25 °C, in good agreement with our experiments (9% mass uptake at 140 psi, 25 °C). In addition, it was found that the FM density increased from $0.3 \, mol \, L^{-1}$ in bulk FM to $3 \, mol \, L^{-1}$ in the confined pores in UiO-66 at 100 psi, 25 °C (Supplementary Fig. 16). We quantified the capillary condensation of FM in UiO-66 by considering the thermodynamics of the equilibrated system using the Two-Phase Thermodynamics Method[51,52]. In particular, we determined the phase change behavior of FM by a novel approach, considering the self-diffusion constant of FM and the number of FM modes that were diffusive (Supplementary Fig. 17). This approach is necessary because while the phase boundaries of a bulk homogenous fluid are given by solutions to the Clausius-Clapeyron equation[53], computational schemes typically rely on locating discontinuities in the relevant thermodynamic functions, such as the molar enthalpy[54]. In the particular case of FM in UiO-66, we found that the molar enthalpy function showed significant uncertainty, especially at low pressures, which obscured unambiguous determination of possible phase boundaries. We found less variability in the calculated self-diffusion constants however, which we used to determine the number of modes of FM that were diffusive.

Supplementary Figure 17a–d shows the translational-diffusion coefficient of bulk FM and adsorbed FM systems. In adsorbed FM systems, we find that as the pressure increases, the translational-diffusion coefficient gradually increases until a certain pressure

(the phase transition point), after which the translational-diffusion coefficient monotonically decreases with increasing pressure. The reduced intermolecular distances between (gaseous) FM molecules before the transition pressure results in weaker attractive forces and an increase in the translational-diffusion coefficient. After the phase transition, the compressed, liquefied FM molecules experiences reduced translational degrees of freedom, and are less diffusive. It is shown that the transition conditions (pressures/translational-diffusion coefficients) of adsorbed FM models at 243, 273, and 298 K occurred at about $10.8 \, psi/2.0·10^{-5} \, cm^2 \, s^{-1}$, $21.7–43.5 \, psi/2.1·10^{-5}–2.3·10^{-5} \, cm^2 \, s^{-1}$, and $103–200 \, psi/1.9·10^{-5}–2.4·10^{-5} \, cm^2 \, s^{-1}$, respectively. It is noted that at 25 °C, the decrease in the translational-diffusion coefficient after the phase transition pressure is not apparent, due to the fact that the FM is approaching its critical properties. The Fig. 3f plots the phase transitions points for bulk FM, compared to adsorbed FM in UiO-66. We find that FM experienced a phase transition at significantly lower pressures in UiO-66, compared to the bulk fluid. In particular, capillary condensation in UiO-66 resulted in liquefied FM at an approximated pressure of 44 psi compared to NIST value of 296 psi in the bulk tank at 0 °C, and at an approximated pressure of 11 psi compared to 118 psi at −30 °C, respectively.

**Electrochemical properties of MPM in liquefied FM solution.** To further investigate the electrochemical properties of cells employing the MPM with FM, the ionic conductivity was measured by a customized two-electrode conductivity cell. To confirm the reliability of our setups, the ionic conductivity of conventional liquid electrolytes at wide temperatures were conducted for comparison (Supplementary Fig. 18). As presented in Fig. 4a, the LGE steadily maintained good conductivity from −60 °C to 30 °C. In contrast, the industry-standard liquid electrolyte (e.g., 1 M $LiPF_6$ in ethylene carbonate (EC)/diethyl carbonate (DEC), 1:1 in volume, 1.2 M $LiPF_6$ in EC: ethyl methyl carbonate (EMC), 3:7 by weight) suffered from rapid conductivity fading with decreasing temperature, suggesting the advantage of using LGE in cold conditions. The MPM-confined FM exhibited an ionic conductivity of ~0.14 mS $cm^{-1}$ at −60 °C

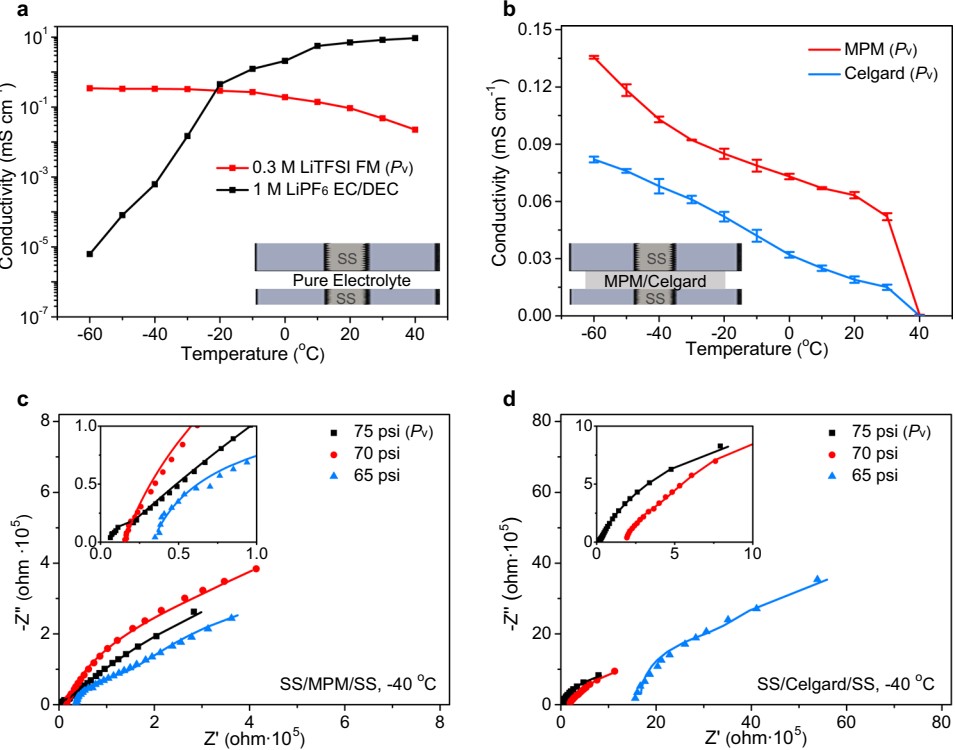

**Fig. 4 Electrochemical properties of MPM-based LGE. a** Ionic conductivity of pure electrolyte with LGE or conventional liquid electrolytes. **b** Ionic conductivity of LGE with the presence of MPM or Celgard-based membranes as the separator at different temperatures. The two symmetric stainless-steel (SS) cells shown in the inset **a** and **b** were set to remain the consistent 500-micron distance. Nyquist impedance of cells with MPM (**c**) or Celgard-based (**d**) membrane as the separator at −40 °C and different FM pressures.

while the Celgard-based system showed less than 0.08 mS cm⁻¹ at the same temperature and pressure (Fig. 4b). This dramatic difference is attributed to the continuous microporous channels formed in the MOF layers and strong affinity between FM and MOFs, which provides improved transport properties.

To probe the nano-confinement effect of the MPM at low temperatures beneath the vapor pressure of the FM electrolyte, a special testing system was designed to control and record the pressure inside the tested cells. The pressure of pure FM at different temperatures was collected, which was fitted with the theoretical data from the NIST database, indicating the viability of the testing system (Supplementary Fig. 19). This setup was then applied to measure the ionic conductivity of the FM-soaked MPM at −40 °C under different pressures. By tuning the cell to 70 psi, 5 psi beneath the vapor pressure of the FM electrolyte, the cells utilizing MPM still offered an electrolyte conductivity of more than 0.02 mS cm⁻¹ at −40 °C (Fig. 4c, Supplementary Table 2), indicating that the MPM retained a reasonable amount of FM inside their pores beneath the bulk vapor pressure, enabling Li⁺ transport. On the contrary, the Celgard-based system produced an ionic conductivity of less than 0.002 mS cm⁻¹ due to inability of Celgard membranes to confine enough FM molecules at such pressure (Fig. 4d, Supplementary Table 2).

**Cell performance with MPM-confined FM electrolyte.** Given the confirmed structural stability, nano-confinement effects, and ionic conductivity results noted above, the MPM was considered as a promising porous membrane for reducing the pressure requirements for LGE. To further verify its application in a real battery system, the MPM was applied to Li//CF$_x$ primary cells, considering the excellent shelf-life and negligible self-discharge behavior of CF$_x$ in conventional liquid electrolytes. To ensure gas

electrolyte transport, a composite cathode was fabricated by introducing 20 wt% of UiO-66 powders to blend CF$_x$ with binder and conductive agent. The corresponding cells were assembled using CF$_x$ composite cathode, Li metal as the anode, and the MPM or Celgard 2400 membranes as the separator (Fig. 5a). To evaluate the cell performance under reduced pressure, the same pressure control system noted before was used for this operation (Supplementary Fig. 19). The MPM-based cell produced an expected high capacity (~855 mAh g⁻¹) at room temperature and vapor pressure, considerably higher than cells with Celgard 2400 membranes (~810 mAh g⁻¹). At −40 °C and vapor pressure, the cells with MPM provided an around 71% room temperature capacity retention (Fig. 5b), which was slightly higher than that of the Celgard membranes. In the same condition, the conventional liquid electrolyte system (1 M LiPF₆ EC/DEC) delivered nearly zero capacity at −40 °C (Supplementary Fig. 20).

More interesting results were found during cell operation at reduced pressure (below $P_v$). To ensure low-pressure gaseous phase operation, the cell pressure was in situ monitored and maintained a nearly constant pressure of 70 psi during discharging operation (Fig. 5c). Notably, the cells with MPM can still maintain 57%, 46%, and 25% of their room temperature capacity (at vapor pressure) at current density of 10, 20, and 40 mA g⁻¹, respectively (Fig. 5d), in spite of the low conductivity of CF$_x$ cathode and reduced charge-transfer kinetics at low temperature. In contrast, the cells with Celgard membrane under the same temperature and pressure produced negligible capacity (~0.03 mAh g⁻¹) at 10 mA g⁻¹. Moreover, during discharging the voltage sharply decreased due to a large internal resistance (Supplementary Fig. 21), suggesting that the large pores of Celgard membrane were not able to retain gas electrolytes at reduced pressure, in agreement with the ionic conductivity measurements.

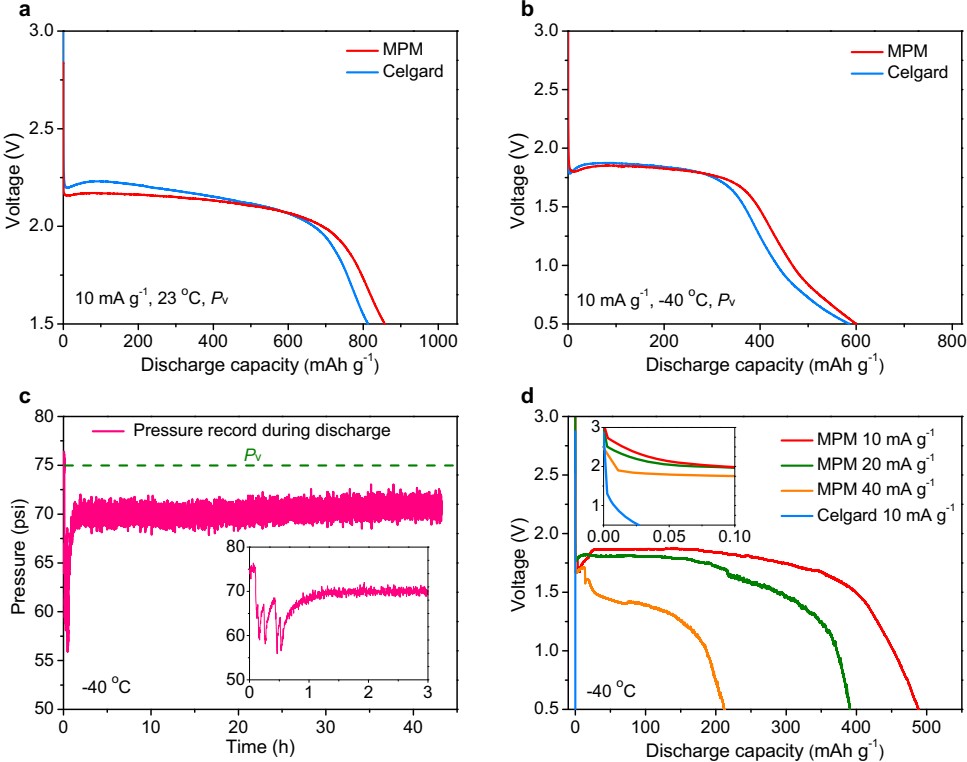

**Fig. 5 Cell performance of Li//CF$_x$ cells.** Discharge characteristics of Li//CF$_x$ cells based on Celgard 2400 or MPM at room temperature (**a**) and −40 °C (**b**) under vapor pressure. **c** In situ monitoring the interior pressure of Li//CF$_x$ cell at −40 °C during the process of discharge. **d** Discharge characteristics of Li//CF$_x$ cells based on Celgard 2400 or MPM at −40 °C, 70 psi and different discharge current density beneath vapor pressure.

## Discussion

To investigate the possible influence of the reduced porosity upon lithiation of the CF$_x$ cathode, the SEM images of the surface and cross-section of CF$_x$ cathode before and after discharge at vapor and low pressure have been compared to examine the influence of the formation of LiF on the electrode porosity. As shown in the Supplementary Fig. 22 and Fig. 23, dense electrode morphology without obvious cracks/pores can be found in all cases, indicating no noticeable change on cathode porosity. In addition, the EIS impedances of Li//MPM//CF$_x$ at −40 °C, 70 psi (lower than vapor pressure) and different depths of discharge (DoDs) have been collected and compared to evaluate whether the expansion of CF$_x$ electrode will render the fade of ionic conductivity during the discharge or not. As presented in Supplementary Fig. 24, the bulk impedances at different DoDs share the relatively small value (50–250 Ω) and do not increase over discharge, which indicates stable ionic conductivity during discharge and thus no noticeable electrolyte leaching from the cathode.

Considering the anode electrode is a solid Li metal, we focused on analyzing the interface between Li metal and LGE confined in MPM to investigate the contact of LGE and the anode. After discharging at −40 °C and vapor pressure (liquid status), the Li//MPM//CF$_x$ cell was disassembled and the surface morphology of Li metal disc was characterized by SEM (Supplementary Fig. 25). The Li metal exhibits homogeneously dispersed pit holes after the stripping process. Similarly, the Li metal after the stripping process at −40 °C and 70 psi also presents even pits despite smaller sizes, indicating a good contact between the LGE and soft Li metal anode retained even under reduced pressure (Supplementary Fig. 26).

To evaluate the stability of CF$_x$ electrode in the LGEs system, the capacity of Li//CF$_x$ cells with MPM-confined LGEs after different storage time were tested, in which no noticeable fade of

discharge capacity was found even after storage for 2 months (Supplementary Fig. 27). The negligible capacity fading suggests the electrochemical compatibility and reasonably good shelf-life of the Li//CF$_x$ cells with MPM-confined LGEs. Note that the slight variation of capacities between cells with 1-, 30-, and 60-days storage time might be due to the variations in cell assembly process, which is often observed in home-made cells. Nevertheless, the above preliminary results together highlight the advantage of MPMs toward confining LGE at reduced pressures for ultralow temperature applications. Typically, shelf-life for 10 or more years will be needed for commercial primary cells, which may be achieved with combination of Li//CF$_x$ cells with MPM-confined LGEs and standard cell structures such as 18,650 cylinders.

In summary, this work shows strong confinement effects of gas molecules in sub-nanometer pores, which leads to significantly increased mass uptake of molecules at pressure lower than the vapor pressure of the bulk counterpart. This was demonstrated by designing "brick-and-mortar"-like MPM membrane as an electrolyte host that consists of dense and continuous subnanometer micropores from MOF building blocks. Both computational and experimental results showed that the electrolyte system based on liquefied FM molecules via capillary condensation in MOFs lowers the equilibrium vapor pressure of FM. The resulted MPM with FM exhibited high structural integrity, decent ion conductivity, and high FM retention, which enabled the operation of high-energy Li cells at low temperatures and reduced working pressure. The unique properties endowed by molecule confinement in nanopores can be extended to design other types of ionic conductive structures for different electrochemical systems, thus opening up new opportunities for emerging applications without sacrificing ease of manufacturability and operation.

## Methods

**Chemicals**. N,N-Dimethylformamide (DMF), Fisher Scientific, 99.9%; ZrCl₄, Alfa Aesar, >99.5%; Cu(NO₃)₂·3H₂O, Alfa Aesar, 99.0%; Terephthalic acid (BDC), Sigma–Aldrich, 98%; Biphenyl-1,4'-dicarboxylic acid (BPDC), Accela Chem Bio Inc., 97.0%; Benzene-1,3,5-tricarboxylic acid (BTC), Alfa Aesar, 98%; Benzoic acid, Sigma–Aldrich, >99.5%; Polyvinylidene difluoride (PVDF), HSV-1800; Poly (vinylidene fluoride-co-hexafluoropropylene), PVDF-HFP 2801; N-Methyl-2-Pyrrolidone (NMP), Sigma–Aldrich, 99.5%; Acetic acid, glacial, Fisher Scientific, 99.7%; Hydrochloric acid (HCl), Fisher Scientific; Fluorinated graphite (CF$_x$), ACS Materials; Lithium bis(trifluoromethane)sulfonimide (LiTFSI), BASF; Fluoromethane (FM), Air Liquide, 99.99%; Tetrahydrofuran (THF), Sigma–Aldrich, >99.9%.

**Preparation of GO@UiO-66**. The graphene oxide (GO, 100 mg) prepared by a modified Hummers' method[55] were dispersed in 100 mL DMF by sonication. The DMF solutions (40 mL) of ZrCl₄ and BDC were added to the above GO dispersion in sequence. After adding 3 mL of acetic acid and stirring at room temperature for 10 min, the mixture was allowed to be heated and stirred at 120 °C for 12 h in an oil bath. After cooling down to room temperature, the gray powder was collected via a centrifugation, and washed with DMF and methanol in sequence (each for three times). The obtained GO@UiO-66 was soaked with methanol overnight, and then was centrifuged and dried under vacuum at 120 °C for 1 day before any fabrication and characterization.

**Preparation of metal-organic framework (MOF)-polymer membranes (MPMs)**. The resulted GO@UiO-66 and PVDF with a mass ratio of 4:1 was dispersed in NMP solution by a Thinky mixer. The obtained slurry was casted on one piece of glass substrate via a doctor blade to control their thickness. After drying in vacuum at 120 °C for 1 day, the membrane was elaborately peeled off from the glass and rolled by roller mill (Durston) to produce the final MPM. The MPM was heated at 120 °C in the glove box under vacuum overnight before performing electrochemical tests.

**Preparation of UiO-66 for soaking tests and UiO-66-based MMMs**. A solution comprised of ZrCl₄ (488 mg), BDC (344 mg), acetic acid (3.6 mL), and DMF (60 mL) were sealed in a Teflon reactor (100 mL) and then heated at 120 °C for 24 h. After cooling down to room temperature, the white powder was collected by a centrifugation, washed with DMF and methanol in sequence, and then dried under vacuum at 120 °C for 1 day before any fabrication and characterization.

**Preparation of UiO-66 for mass change tests and porous additives of cathodes**. A solution comprised of ZrCl₄ (875 mg), BDC (625 mg), acetic acid (50 mL), and DMF (250 mL) was prepared. The solution was then divided into four Teflon reactors (100 mL) and then heated at 120 °C for 24 h. After cooling down to room temperature, the white powder was collected by a centrifugation, washed with DMF and methanol in sequence, and then dried under vacuum at 120 °C for 1 day before any fabrication and characterization.

**Preparation of HKUST-1**. Typically, the BTC (1 g) and Cu(NO₃)₂·3H₂O (2 g) were added into a mixture solution with DMF, H₂O, and ethanol (16 mL/16 mL/16 mL) and stirred for 15 min. The resulted blue mixture was then transferred into an oven and was allowed to react at 85 °C for 12 h. After cooling down to room temperature, the blue powder was collected by centrifugation, washed with DMF and methanol in sequence, and then dried under vacuum at 120 °C for 1 day before any fabrication and characterization.

**Preparation of UiO-67 for soaking tests and UiO-67 based MMMs**. The synthesis method of UiO-67 followed the procedure of UiO-66 (for soaking tests and UiO-66-based MMM) except for replacing BDC by BPDC in the same molar ratio.

**Preparation of MOF-808**. Typically, 0.583 g of ZrCl₄ and 0.175 g of BTC was dissolved in 25 mL of DMF. Subsequently, 14 mL of acetic acid was added. After stirring for 5 min, the mixture was allowed to be heated at 135 °C for 24 h. The obtained powder was collected by centrifuged (washed with DMF for three times and then methanol for three times) and soaked in methanol for 1 day. Then the powder was dried under vacuum at 120 °C for 1 day before any fabrication and characterization.

**Preparation of MOF-based mixed matrix membranes (MMMs)**. An acetone solution (8 mL) of MOF powders (0.3 g) and an NMP solution (0.7 mL) of PVDF-HFP (0.15 g) were prepared by sonication. The above solutions were mixed, vigorously shaken, and then under further sonication for 30 min. The mixture was transferred into a vacuum oven (~0.25 atm) for 40 min to remove the acetone solvent. Subsequently, the concentrated suspension was further homogenized by a Thinky mixer for 10 min. The resulted slurry was then poured onto a clean glass substrate, casted by a doctor blade to control the thickness (300–400 µm). The

sample was dried at ambient pressure (60 °C) first and then under vacuum (120 °C). The MMMs were easy to be peeled off from the substrate upon wetting with ethanol. The free-standing MMMs were heated at 120 °C in the glove box in vacuum overnight before electrochemical tests.

**Fabrication of CF$_x$ cathodes**. Typically, 240 mg of CF$_x$ was sufficiently mixed with 80 mg of UiO-66 and 40 mg of carbon black in the mortar by hand. The NMP solution containing 40 mg of PVDF was dropped into the above powder mixture. The homogeneous slurry obtained by a Thinky mixer was casted on clean Al foil. The electrode was dried under vacuum at 120 °C overnight before any use.

**Electrochemical measurements**. The ionic conductivity of the pure electrolyte of 0.3 M LiTFSI in FM and the selected electrolyte soaked Celgard membrane, MMM and MPM were measured by a customized two-electrode (Stainless Steel 316 L) conductivity cell. The electrolyte composition was 0.3 M LiTFSI in FM with excessive LiTFSI to ensure there was sufficient salt left after reaching the pressure below vapor pressure at the set temperature. It's noteworthy the 0.3 M LiTFSI 0.3 M THF in CO₂, FM (THF:CO₂:FM = 1:4:95 vol%) was only used to measure the ionic conductivity of MMM for choosing the MOF candidates. The cell constant was frequently calibrated by using OAKTON standard conductivity solutions at 0.447, 1.5, 15, and 80 mS·cm⁻¹, respectively. A constant thickness spacer was positioned between the two electrodes, which ensured no obvious distance changes during multiple-time measurements. The electrolytic conductivity value was obtained with a floating AC signal at a frequency determined by the phase angle minima given by the electrochemical impedance spectroscopy (EIS) using the following equation:

$$\sigma = KR^{-Q} \tag{2}$$

where $R$ is the tested impedance (Ω), $K$ the cell constant (cm⁻¹) and $Q$ the fitting parameter[56]. All of data acquisition and output were done by LabView Software, which was also used to control an ESPEC BTX-475 programming temperature chamber to maintain the cell at a set temperature in 30 min intervals.

EIS measurements were conducted with a sinusoidal probe voltage of 5 mV from 1 mHz to 1 MHz in a customized high-pressure stainless-steel cell. The spectra were fitted by an equivalent circuit model using ZView software.

The pressure tuning process was finished by the customized tuning system, as shown in Supplementary Fig. 19. It consisted of one SPT25-10-1000A ProSense pressure transmitter with 0–1000 psi range, four Swagelok ball valves, one customized high-pressure cell, one Swagelok tube fitting and several PTFE thermal-resistant tubes. The pressure transmitter was connected to the customized cell through a T-sized Swagelok tube fitting of which the other side was connected to an Edwards vacuum pump using PTFE tubes. The pressure transmitter was opened during the whole experiment period to monitor the pressure change and the testing temperature was controlled by ESPEC BTX-475 programming temperature chamber to maintain the cell at a set temperature. All of the data transfer was controlled by the LabView program. During tuning process, the valve C and D was kept open and after the tuning process, valve C was close and valve D was kept open to record the pressure value during the discharge process. We decreased the pressure from vapor pressure to the pressure below vapor pressure at the set temperature controlled by valve B. When the pressure was stable at the set pressure at −40 °C, cells were tested by a Biologic SP-150 electrochemistry workstation.

Battery test under vapor pressure was performed by an Arbin battery test station (BT2043, Arbin Instruments, USA) and battery test under reduced pressure was performed by a Biologic SP-150 electrochemistry workstation in custom designed high-pressure stainless-steel coin cells, with Li metal (FMC Lithium, 1 mm thickness, ³/₈ inch diameter) as the counter electrode and CF$_x$ (¹/₄ inch diameter) mixed with 20 wt% of UiO-66 as the working electrode. 5% CO₂ in FM was added to stabilize the lithium metal[15]. All of the electrochemical test and ionic conductivity test have included the CO₂ to keep consistent test system. Three layers of porous polypropylene separator (Celgard 2400, 25 µm) or one layer of MPM (~75 µm) was applied for all the electrochemical experiments.

**Material characterizations**. Morphologies of various MOF powders and MOF membranes were detected by Scanning Electron Microscope (FEI XL30, UHR-SEM). Powder X-ray diffraction (XRD) patterns were measured on Bruker D2 Phaser (Germany) under Cu Kα radiation. N₂ adsorption/desorption isotherms were tested with Quantachrome instrument (QUADRASORB SI, American) at 77 K. The model for pore size distribution simulation was NLDFT. Degassing condition was set as 120 °C for 12 h. Fourier transform-Infrared (FT-IR) spectra of different MOF powders were carried on Nicolet 6700 with Smart-iTR diamond ATR crystal using attenuated total reflectance mode. The spectrometer was equipped with a liquid nitrogen-cooled MCT-A detector to receive the IR signal from 600 to 4000 cm⁻¹. Raman spectra of liquefied gas electrolytes were carried on Renishaw inVia confocal Raman microscope with an excitation wavelength of 532 nm calibrated with Si (520 nm) before test.

X-ray nano-computed tomography (Nano-CT) test was performed on the MPM sample, which was individually punched into a film with a radius of 2 mm and placed within a PTFE cylindrical tube with PTFE rod ends to provide sealing. The scan was conducted using a ZEISS Xradia 510 Versa micro-CT instrument

with a voxel size of 75 μm and an exposure of 10 s. An X-ray energy of 80 keV with a current of 87.6 μA was used to collect 1801 projections with a high-energy filter at a ×4 magnification. The final reconstruction of the raw scan data was performed with a beam hardening constant of 0 and a center shift constant of −2.9 using software provided by ZEISS. Post measurement analysis was performed by Amira-Avizo using the Despeckle, Deblur, and Delineate modules for data sharpening and filtration provided by the software. The volume fraction and area of the MOF structure were determined with the Materials Analysis and Volume Fraction modules within Amira-Avizo.

**Mass change test**. Mass change test was done by soaking different MOF powders into a customized high-pressure stainless-steel cell filled with FM under different pressure for three days at room temperature. After the soaking process, most of FM was firstly purged out of the cell and then the cell was transferred into an Ar-filled glove box to measure the total mass of soaked MOFs powders using a scientific scale with $10^{-4}$ g accuracy. Subsequently, the MOF powders were rest and the mass was recorded. The normalized mass change is the mass of MOF powders after soaking process divided by the mass of UiO-66 before soaking process. It's noticeable that there was less than 5% mass loss during sample transfer process.

**Compatibility test of MOFs in FM**. Stability tests for different MOF powders in FM were carried out at vapor pressure (500 psi) at room temperature for 3 days with excessive liquid FM immersing MOF particles. After the soaking process, BET, XRD, FT-IR, and Raman analysis were done on the soaked MOF powders. As for the FT-IR and Raman, the exposed time of tested samples to the air was less than 30 s.

Stability tests for the MPMs and MMMs in selected electrolytes were carried out at vapor pressure (500 psi) at room temperature for 3 days with excessive 0.3 M LiTFSI in FM electrolytes. After the soaking process, the soaked samples were conducted the SEM analysis compared with the pristine films.

**Computational details**. The simulation parameters were described in Supplementary Table 1, where the FM properties were obtained via QM calculations at the MP2/aug-cc-pVTZ level of theory using the Q-Chem 5.0 electronic structure package[46]. The UiO-66 structure was initially optimized via MD approach from LAMMPS software[47] with the starting structure shown in Supplementary Fig. 14, and the procedure was detailed in the following "MD of FM/UiO-66 and UiO-66 systems" section. Further, GCMC simulations (the MCCCS Towhee simulation package[48]) were applied to model the molecules' loading value inside the optimized UiO-66 structure. GCMC is a procedure involving insertion/deletion molecules between a system and a reservoir to eventually make system/reservoir in thermodynamic equilibrium, under conditions of constant chemical potential (μ), volume (V) and temperature (T). For each GCMC computation, 3 million moves were performed, and we tested that convergence was obtained in each simulation. The initial 2 million moves were used to equilibrate the system, while the last 1 million moves were used to obtain the relevant statistics and absorption capacities. Besides the FM/UiO-66 models, the adsorption isotherms of $CH_4$ and $CO_2$ inside UiO-66 were also calculated in order to confirm the accuracy of our GCMC approach, as shown in Supplementary Fig. 15[49,50]. After the adsorption capacity of FM were determined, the final snapshot of the system was used as input for further MD simulations.

**MD simulations of FM/UiO-66 and UiO-66 systems**. We initiated each MD simulation with a geometry optimization at 0 K, comprising 500 steps of conjugated gradient (CG) minimization (cell coordinates and atom positions) followed by 10 ps of canonical (constant particles, volume and temperature or NVT) dynamics to heat the system from 1 K to the defined temperature. This was followed by isothermal iso-baric (constant particles, pressure, and temperature or NPT) dynamics for 1 ns at the required pressures, followed another 0.8 ns NVT dynamics to further equilibrate the system. The last 0.2 ns NVT data were used to collect thermodynamic statistics and to analyze the system properties. In all our MD simulations, the temperature damping constant was 0.1 ps, and the pressure damping constant was 2.0 ps. The equations of motion used are those of Shinoda et al.[57], which combine the hydrostatic equations of Martyna et al.[58] with the strain energy proposed by Parrinello and Rahman[59]. The time integration schemes closely follow the time-reversible measure preserving Verlet integrators derived by Tuckerman et al.[60].

**MD of bulk FM systems**. For bulk FM, a simulation cell comprising 216 molecules at the 298 K saturated vapor density 2.7688 mol L$^{-1}$ from NIST database) was subjected to 500 steps of CG minimization, followed by 10 ps of Langevin dynamics to heat the system from 1 K to a defined temperature. This was followed by 1 ns of NPT dynamics, and a further 5.5 ns of Langevin dynamics to properly equilibrate the system. Longer simulation times, compared to FM/UiO-66 and UiO-66 systems, and the application of Langevin dynamic were to ensure thermal equipartition of the energy, due to the low density of the bulk FM system. After equilibration, we ran a further 0.7 ns of NVT dynamics, with statistics collected during the last 0.2 ps used to analyze the system properties. The temperature and pressure damping constants were the same as above.

**Self-diffusion constant**. The self-diffusion constant $D$ was obtained using the Green-Kubo VAC formulism in linear response theory:[54]

$$D = \frac{1}{N}\sum_{1}^{N}\int_{0}^{\infty}\langle v_i(t)v_i(0)\rangle dt \qquad (3)$$

where $t$ is time, $v_i$ the axial COM velocity of molecule $i$ and the brackets denote an autocorrelation that is summed over all molecules. These calculations were obtained from additional simulations in the NVT ensemble, after pressure equilibration has been achieved. Snapshots of the system (atomic coordinates and velocities) were saved every 1 fs during a 0.5 ns simulation. Statistical averaging was performed by using 10 windows of 50 ps each. Our previous work has shown that trajectory windows of 50 ps were long enough to have converged self-diffusion constants by this approach[52].

## Data availability
The data that support the findings of this study are available within this article and Supporting information, or from the corresponding author upon reasonable request.

## Code availability
An in-house code that implements the 2PT method is available from the authors or online at https://github.com/atlas-nano/2PT.

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

## Acknowledgements

This work was supported by an Early Career Faculty grant from NASA's Space Technology Research Grants Program (ECF 80NSSC18K1512) to Z.C. This research was partially supported by NSF through the UC San Diego Materials Research Science and Engineering Center (UCSD MRSEC), DMR-2011924. Both Z.C. and T.P. acknowledge the start-up fund from the Jacob School of Engineering at the University of California, San Diego (UCSD). M.M. and Y.S.M acknowledge the partial funding support from the subcontract by Sandia Laboratory Directed Research and Development (LDRD) project 218253. We would also like to acknowledge the National Center for Microscopy and Imaging Research (NCMIR) technologies and instrumentation supported by grant R24GM137204 from the National Institute of General Medical Sciences for the nano-CT work. A portion of cell fabrication and electrochemical testing work was performed in the UCSD-MTI Battery Fabrication and the UCSD-Arbin Battery Testing Facility. This work was performed in part at the San Diego Nanotechnology Infrastructure (SDNI) of UCSD, a member of the National Nanotechnology Coordinated Infrastructure, which is supported by the National Science Foundation (Grant ECCS-1542148). This research used resources of the National Energy Research Scientific Computing Center, a DOE Office of Science User Facility supported by the Office of Science of the U.S. Department of Energy under Contract No. DE-AC02-05CH11231. This work also used the Extreme Science and Engineering Discovery Environment (XSEDE), and the Comet and Expanse supercomputers at the San Diego Supercomputing Center, which is supported by National Science Foundation grant number ACI-1548562.

## Author contributions

Z.C. conceived the idea and supervised the project. G.C. designed the MPM., Y.Y., D.X., and G.C. performed the experiments and collected the data. T.A.P. and A.A.C. conducted the computer simulations. J.H., Y.Y. and Y.S.M. discussed the results and commented on the manuscript. J.S. carried out the nano-CT experiments. K.H.K. and T.H.H. designed and prepared the GO templates. M.L. performed the SEM experiments. D.M.D. and M.M. assisted in performing low-temperature testing. G.C., Y.Y. and D.X. co-wrote the paper with the input from all authors. These authors contributed equally to this work: G.C., Y.Y. and D.X.

## Competing interests

The authors declare no competing interests.
