## [Peer Review File · Nature Communications]

REVIEWER COMMENTS

Reviewer #1 (Remarks to the Author):

Liquefied gas electrolytes are promising to extend the low temperature operational capability of batteries due to their low freezing point, but pose a safety risk due to their high vapor pressure. The manuscript describes capillary condensation of gas electrolyte by strong confinement in sub-nanometer pores of metal-organic framework (MOFs). Using a "brick-and-mortar"-like MOF-polymer membrane (MPM) as an electrolyte host that consists of dense and continuous sub-nanometer micropores from MOF building blocks, it is shown experimentally and computationally that the capillary condensation in MOFs lowers the equilibrium vapor pressure of fluoro-methane FM. The MPM with FM has good structural integrity, decent ion conductivity at low temperature with high FM retention, which extended Li-CFx primary cell to operate at extremely low temperatures with low working pressure. This work is sufficiently novel, looks interesting from applications perspective, and especially promising for low temperature electrochemical devices. I therefore recommend publication of this paper in the journal, but only after revising the manuscript by addressing the following comments.

1. Upon exposure to a warm temperature, the FM condensed in the pores will come out. How fast will the electrolyte recondense into the MOF. In other words, how fast is the condensation kinetics?
2. Page 3 line 36: It is probably worth mentioning that Li-SOCl₂ cell can function down to -80°C.
3. Page 4 Line 53: Apart from the pore size, can the liquid/vapor surface tension be changed by any chemical modification of MOF?
4. The condensation, I believe, is the electrolyte with FM as solvent. What is the salt used here and at what is the concentration?
5. The CFX cathode expands during discharge with the formation of LiF. Will it cause a reduction in the cathode porosity and the electrolyte in the composite cathode to be expelled from the MOFs?
6. The LGE in the cathode and separator will be in liquid state due to capillary forces, but isn't the electrolyte at the anode, i.e., in contact with the anode in the gaseous form?
7. Page 6 line 106: What is the Celgard that is being compared here? Is it Celgard with LGE absorbed within (absorbs less electrolyte and hence has lower conductivity) or Celgard with a conventional liquid electrolyte?
8. What is the proportion % of the electrolyte in the MOFs? Since some portion of the electrolyte in the liquid state within the MOF, the rest of the electrolyte will be in the gaseous state and will determine the pressure, correct?
9. The conductivities of typical liquid electrolytes at these low temperatures need to be included here for comparison.
10. Page line 152: Stability over 3 days is grossly inadequate, the battery will need to survive for years (> 10 years for conventional Li-CFx). Any comment on the long-term effects or on the projected shelf life of the batteries with LGE?
11. Line 190. What is the optimum uptake of FM to provide sufficient conductivity? 140 psi is fairly high at room temperature and requires a thick cell casing (still unsafe), which will reduce the specific energy.
12. Cathode is in contact with MOF but the electrolyte is contained within the pores of MOF. How well good cathode/liquid electrolyte interface be developed?
13. Line 201: What are the values for the diffusion coefficient?
14. Line 248: The capacity of 940 mAh/g is rather high for a CFX cathode. Its theoretical capacity is 865 mAh/g.

Reviewer #2 (Remarks to the Author):

This is a very nice paper demonstrating the confinement of a liquid gas electrolyte in a MOF membrane where the MOF membrane has been fabricated into reinforced, continuous materials compositing PVDF with nano sheets of MOF grown directly onto a GO support. LGE is a relatively new concept that one of the authors has demonstrated recently, however the use of nanoconfinement in MOF membranes is novel as far as I can tell. The performance of the LGE

based on a fluorinated methane and LiTFSI shows excellent low temperature conductivity even under lower pressure, especially when compared with the same LGE supported on Celgard. The battery performance of these mixed membrane electrolytes is very impressive relative to a comparison with the current organic liquid electrolyte which hardly cycles at the low temperatures desired in this work. The comparison of device performance between the 'brick and mortar' MOF/GO membrane and the Celgard shows a slightly better performance but I would have liked to see some discussion of reproducibility here as the difference is not really that significant. There is quite a lot of discussion relating to simulations and calculations in the experimental section but I don't see the data from this described in any detail in the manuscript. Perhaps this could be elaborated.

Rebuttal Key

Blue text – comment responses

Highlighted text – newly edited/added text in the revised manuscript

Boxed text & figures - Excerpts from the newly revised manuscript/SI

Reviewer 1

Overall comments: *“Liquefied gas electrolytes are promising to extend the low temperature operational capability of batteries due to their low freezing point, but pose a safety risk due to their high vapor pressure. The manuscript describes capillary condensation of gas electrolyte by strong confinement in sub-nanometer pores of metal-organic framework (MOFs). Using a “brick-and-mortar”-like MOF-polymer membrane (MPM) as an electrolyte host that consists of dense and continuous sub-nanometer micropores from MOF building blocks, it is shown experimentally and computationally that the capillary condensation in MOFs lowers the equilibrium vapor pressure of fluoro-methane FM. The MPM with FM has good structural integrity, decent ion conductivity at low temperature with high FM retention, which extended Li-CFX primary cell to operate at extremely low temperatures with low working pressure. This work is sufficiently novel, looks interesting from applications perspective, and especially promising for low temperature electrochemical devices. I therefore recommend publication of this paper in the journal, but only after revising the manuscript by addressing the following comments.”*

Response: We thank the review for the assessment on the novelty of our work and the promise of our designed electrolyte systems for low-temperature electrochemical devices. We have carefully designed and performed new experiment to fully addressed the specific technical questions detailed below.

Comment #1: *“Upon exposure to a warm temperature, the FM condensed in the pores will come out. How fast will the electrolyte recondense into the MOF. In other words, how fast is the condensation kinetics?”*

Response: Thanks for the insightful concerns. In order to evaluate the recondensation rate into the MOF pores, we have recorded the impedance change of MPM confined liquefied gas electrolyte (LGE) at different temperatures with the fixed amount of fluoromethane (FM) gas in a sealed cell. Typically, a fixed volume of FM gas was filled in the stainless steel (SS)//MPM//SS cells through controlling the pressure in the cell to 70 psi (lower than vapor pressure) at -40 °C, followed by sealing the cell. The EIS impedance at this condition was then collected as the initial reference. Subsequently, the temperature was increased to 20 °C and then hold at this temperature for another 1 hour. Finally, the temperature was reduced to -40 °C, followed by the monitoring the impedance

changes at different standing times. As shown in **Fig. R1**, it can be observed that the impedance value immediately increases upon exposure to elevated temperature (20 °C) compared with the initial value at -40 °C, due to the evaporation of MPM confined liquified gas at elevated temperature, which agrees with the reviewer’s hypothesis. Nevertheless, the impedance quickly drops to near the original value within 16 min after tuning back to the initial temperature (-40 °C), attributed to the redissolution of Li salts enabled by the recondensation of gas electrolyte at reduced temperature, thereby promoting facile diffusion of Li⁺. Furthermore, the small amount of residual resistance can be fully eliminated during the subsequent 2.5 hours holding, which enables fully dissolution and wetting of salts in/around all the MOF pores. Overall, it takes about 3 hours to allow complete gas recondensation, salt re-dissolution and full wetting of the MPM with LGEs.

Comment #2: “Page 3 line 36: It is probably worth mentioning that Li-SOCl₂ cell can function down to -80°C.”

Response: We appreciate the thoughtful suggestion. Some literatures about low-temperature Li-SOCl₂ cells have been cited (see ref. 12-14 in the main text, page 27). Corresponding description about this has also been added in the main text (see page 3, line 36-38).

Main text: page 3

The state-of-the-art lithium-ion batteries (LIBs) are mostly restricted to perform in mild conditions due to the drastically decreased ionic conductivity and increased charge transfer impendence of electrode/electrolyte interfaces at ultra-low temperatures,¹⁻¹¹ despite that some cells like lithium-thionyl chloride batteries are capable of operation down to -80 °C for low power applications.¹²⁻¹⁴

Main text reference section: page 27

12: West, W. C. *et al.* Sulfuryl and thionyl halide-based ultralow temperature primary batteries. *J. Electrochem. Soc.* **157**, A571-A577 (2010).

13: Hills, A. J. & Hampson, N. The Li-SOCl₂ cell—a review. *J. Power Sources* **24**, 253-271 (1988).

14: Schlaikjer, C. R., Goebel, F. & Marincic, N. Discharge reaction mechanisms in Li/SOCl₂ cells. *J. Electrochem. Soc.* **126**, 513-522 (1979).

Comment #3: “Apart from the pore size, can the liquid/vapor surface tension be changed by any chemical modification of MOF?”

Response: We are thankful for the review’s insightful comment. To investigate if the liquid and vapor surface tension would be influenced by the chemical modification of MOFs, we also synthesized similar MOF materials with different pore structure and functional group, including UiO-66-NO₂ (a UiO-66 analogue with additional -NO₂ functional group), and UiO-67 (a UiO-66 analogue with extended linkers). The mass change test of FM soaked UiO-66-NO₂ and UiO-67 have been conducted and compared with pristine UiO-66 system. As presented in **Supplementary Fig. 13**, UiO-66-NO₂ with a polar group on the MOF skeletons exhibits a higher uptake capacity to confine the FM gas and slower release rate, while UiO-67 with increased pore sizes compared with UiO-66 poses reduced retention times due to weak nanoconfinement effect. The above results indicate that the liquid/vapor surface tension can be adjusted by modify the nanoscale environments of MOF pores. More systematic investigation on adjusting liquid/vapor surface tension through chemical modification of MOF pore environments are undergoing and will be presented in our future works. Corresponding description has been added in the main text (See page 9 line 182-186 and page 10 line 187-188)

Supporting information: page 14

Supplementary Figure 13 | Mass change tests of liquified FM soaked UiO-66 and its analogues.

Main text: page 9 and 10

As shown in Fig. 3c, the mass of the UiO-66 powders increased by ~12% after soaking at ~500 psi, demonstrating the ability of UiO-66 to store a large volume of liquified FM molecules (corresponding to molar ratio of FM:UiO-66 at 5.7:1 for the absorbed sample). It is worth noting that the liquid/vapor surface tension can be further changed by the chemical modification of the MOF skeleton. As presented in Supplementary Fig. 13, UiO-66-NO₂ (a UiO-66 analogue with additional -NO₂ functional group) with a polar group on the MOF skeletons exhibits a little bit higher uptake capacity to confine the FM gas and slower release rate, while UiO-67 (a UiO-66 analogues with extended linkers) with increased pore sizes compared with UiO-66 poses reduced retention times due to weak nanoconfinement effect. Considering the high complexity for simulating a variety of pore structure and chemical moieties, we select UiO-66 as the model system.

Comment #4: “The condensation, I believe, is the electrolyte with FM as solvent. What is the salt used here and at what is the concentration?”

Response: Thanks for pointing this out. The bis(trifluoromethylsulfonyl)amine lithium (LiTFSI) salt is utilized as the lithium salt and its concentration of LiTFSI is fixed at 0.3 mol per liter in FM. The detailed information is also showed in experiential procedures (Page 19 Line 404).

Comment #5: “The CF_x cathode expands during discharge with the formation of LiF. Will it cause a reduction in the cathode porosity and the electrolyte in the composite cathode to be expelled from the MOFs?”

Response: We appreciate the insightful concerns. The SEM images of the surface and cross-section of CF_x cathodes before and after discharge at vapor and reduced pressure have been compared to study the influence of the formation of LiF on the electrode porosity. As shown in the **Supplementary Figures 22 and 23**, dense electrode morphology without obvious cracks can be found in all cases, indicating no change on porosity. In addition, the EIS impedances of Li//MPM// CF_x at $-40\text{ }^\circ\text{C}$, 70 psi (lower than vapor pressure) and different depths of discharge (DoDs) have been collected and compared to evaluate whether the explanation of CF_x electrode will render the fade of ionic conductivity during the discharge or not. As presented in **Supplementary Figure 24**, the bulk impedances at different DoDs share the relatively small value ($50\ \Omega \sim 250\ \Omega$) and do not increase over discharge, which indicates stable ionic conductivity during discharge and thus no noticeable electrolyte leaching from the cathode. Additionally, the UiO-66 we chose is the one with the highest modulus in common MOFs, which is close to many ceramics (*J. Phys. Chem. Lett.* 2013, 4, 6, 925-930). Therefore, we believe it is unlikely that outside particles can exert any influence, *i.e.*, expelling or squeezing the electrolytes in MOFs.

Supplementary Figure 23 | Cross-sectional SEM images of CF_x electrodes with 20 wt.% of UiO-66. (a, b) the pristine electrode; (c, d) the electrode after discharge at $-40\text{ }^\circ\text{C}$ and vapor pressure; (e, f) the electrode after discharge at $-40\text{ }^\circ\text{C}$ and 70 psi.

Discussion regarding this question has been added in the revised manuscript on page 14.

Main text: page 14

To investigate the possible influence of reduced porosity upon lithiation of the CF_x cathode, the SEM images of the surface and cross-section of CF_x cathode before and after discharge at vapor and low pressure have been compared to examine the influence of the formation of LiF on the electrode porosity. As shown in the Supplementary Figures 22 and 23, dense electrode morphology without obvious cracks can be found in all cases, indicating no noticeable change on cathode porosity. In addition, the EIS impedances of Li//MPM// CF_x at $-40\text{ }^\circ\text{C}$, 70 psi (lower than vapor pressure) and different depths of discharge (DoDs) have been collected and compared to evaluate whether the expansion of CF_x electrode will render the fade of ionic conductivity during the discharge or not. As presented in Supplementary Fig. 24, the bulk impedances at different DoDs share the relatively small value ($50\ \Omega \sim 250\ \Omega$) and do not increase over discharge, which indicates stable ionic conductivity during discharge and thus no noticeable electrolyte leaching from the cathode.

Supplementary Figure 24 | Nyquist impedance of Li//CF_x cell (with 20 wt.% of UiO-66 in the cathode) using MPM at 70 psi, -40 °C and different DoD. Inset shows the detailed comparison of bulk impedances at high frequency regions.

Comment #6: “The LGE in the cathode and separator will be in liquid state due to capillary forces, but isn’t the electrolyte at the anode, i.e., in contact with the anode in the gaseous form?”

Response: We thank the reviewer for the constructive comments. We believe that the LGE retains a liquid or condensed state at the anode/electrolyte interface like in the cases of the cathode and separator. Considering the anode electrode is a solid Li metal, we focused on analyzing the interface between Li metal and LGE confined in MPM to investigate the contact of LGE and the anode. After discharging at -40 °C and *vapor pressure (liquid status)*, the Li//MPM//CF_x cell was disassembled and the surface morphology of Li metal disc was characterized by SEM (**Supplementary Fig. 25**). The Li metal exhibits homogeneously dispersed pit holes after stripping process. Similarly, the Li metal after the stripping process at -40 °C and *70 psi also* presents even pits despite smaller sizes, indicating a good contact of liquid LGE and soft Li metal anode retained even under reduced pressure (**Supplementary Fig. 26**). The following discussion has been added in the revised manuscript (page 15):

Main text: page 15

Considering the anode electrode is a solid Li metal, we focused on analyzing the interface between Li metal and LGE confined in MPM to investigate the contact of LGE and the anode. After discharging at -40 °C and *vapor pressure (liquid status)*, the Li//MPM//CF_x cell was disassembled and the surface morphology of Li metal disc was characterized by SEM (**Supplementary Fig. 25**). The Li metal exhibits homogeneously dispersed pit holes after stripping process. Similarly, the Li metal after the stripping process at -40 °C and *70 psi also* presents even pits despite smaller sizes, indicating a good contact between LGE and soft Li metal anode retained even under reduced pressure (**Supplementary Fig. 26**).

Supplementary Figure 25 | Characterization of stripped Li metal under vapor pressure (liquid state). (a) Schematic showing the of Li//CF_x cell with a relatively large Li chip as the anode while a small CF_x electrode disc as the cathode. (b) SEM image of the Li metal anode obtained from disassembling the Li//MPM//CF_x cell after discharging at -40 °C and vapor pressure. The white dotted line indicates the boundary between stripped and unreacted Li metal. The enlarged SEM images of (c) stripped and (d) unreacted Li metal.

Supplementary Figure 26 | Characterization of stripped Li metal under reduce pressure. (a) Schematic showing the Li//CF_x cell with a relatively large Li chip as the anode while a small CF_x electrode disc as the cathode. (b) SEM image of the Li metal anode obtained from disassembling the Li//MPM//CF_x cell after discharging at -40 °C, and 70 psi. The white dotted line indicates the interface of stripped and unreacted Li metal. The enlarged SEM images of (c) stripped and (d) unreacted Li metal.

Comment #7: “Page 6 line 106: What is the Celgard that is being compared here? Is it Celgard with LGE absorbed within (absorbs less electrolyte and hence has lower conductivity) or Celgard with a conventional liquid electrolyte?”

Response: We thank the reviewer for the careful concern. The Celgard we used here is Celgard 2400. We compared the ionic conductivity of Celgard with both LGE and conventional liquid electrolyte. It should be noted that the Celgard membrane with conventional liquid electrolyte (e.g., 1 M LiPF₆ in EC:DEC 1/1 by vol.) cannot work at extremely low temperature (e.g., < -30 °C) due to the freezing of liquid electrolyte. To further clarify the experiment details, the following description has been added in the main text (Page 6 line 110).

Main text: page 6

It was revealed that among various MOFs, UiO-66 and UiO-67 based MMMs provided the highest ion conductivity. Of note, at -60 °C the UiO-66 MMM exhibited an ionic conductivity of 0.67 mS/cm while UiO-67 exhibited 0.75 mS/cm, higher than that of Celgard 2400 (0.36 mS/cm) with LGE.

Comment #8: “What is the proportion % of the electrolyte in the MOFs? Since some portion of the electrolyte in the liquid state within the MOF, the rest of the electrolyte will be in the gaseous state and will determine the pressure, correct?”

Response: We thank the reviewer for this insightful question. Taking our setup at -70 psi (482633 Pa) and -40 °C (233.15 K) as the example, the molar number of the gaseous FM is roughly calculated based on the formula of $P \cdot V = n \cdot R \cdot T$. (Note that the compressibility factor of FM is 0.993, so it is fair to use the ideal gas law for simplicity). Considering the volume ($2.7 \cdot 10^{-5} \text{ m}^3$) of our dog-bone cell setup is much larger in comparison with the relatively small volume of the electrode and separator and so on, the volume of the later can be ignored, thereby $n = \frac{P \cdot V}{R \cdot T} = \frac{482633 \cdot 2.7 \cdot 10^{-5}}{8.314 \cdot 233.15} = 6.72 \cdot 10^{-3} \text{ mol}$. We can calculate the mass of gas state FM is 0.23 g. On the other hand, the mass of FM confined in MOFs is 0.038g measured by the following equation:

$$m_{\text{condensed FM}} = P_{\text{MPM}} V_{\text{MPM}} \rho_{\text{adsFM}}$$

$m_{\text{condensed FM}}$: mass of condensed FM in MOFs (achieved by the MD simulation); P_{MPM} : porosity of MPM (26.5%); V_{MPM} : total volume of MPM (0.708 cm^3); ρ_{adsFM} : density of adsorbed FM in MOFs ($\sim 0.204 \text{ g cm}^{-3}$).

Therefore, the FM proportion of condensed FM in MOFs is around 16.5%. We would like to point out that this calculation is based on our home-built cells with extra size for the purpose of easy control of gas feeding. It also worth noting that the proportion of LGE in MOFs will highly depend on other variables, including the structure of cells, the working temperature, and the amount of filled gas, etc. In the industrial 18650-cell form factor (*Adv. Mater.* 2019, 9, 1803170), a major proportion ($\sim 92.2\%$) of electrolyte can be expected to be remained in liquid state within the MOFs, because most of the space is occupied by MOFs and electrodes.

As for the second question, after tuning pressure below the vapor pressure, the cell pressure is determined by the gaseous state of FM. To confirm this, various reduced pressures have been adjusted by further purging gas out of the cells. As shown in Fig. R2, the cell pressure will stabilize at the different setting pressures (below vapor) after step-by-step release the gas.

Comment #9: “The conductivities of typical liquid electrolytes at these low temperatures need to be included here for comparison.”

Response: We appreciate the thoughtful suggestion. We have added more common electrolytes’ ionic conductivity data (**Supplementary Fig. 18**), including 1.0 M LiPF₆ in 1:1 EC:DEC by *vol.*, 1.0 M LiPF₆ in 1:1 EC:DMC by *vol.*, 1.2 M LiPF₆ in 3:7 EC:EMC by *wt.*, 1 M LiTFSI in DME:DOL 1:1 by *vol.* and 1.0 M LiBF₄ in 4:1 DME:PC by *vol.*, which is consistent with literatures reported value (*Nano Lett.* 2019, 19, 8664-8672; *Angew. Chem. Int. Ed.* 2019, 58, 18892-18897; *J. Electrochem. Soc.* 2017, 164, A3109-A3116). It should be noted that conventional liquid carbonate electrolytes will be frozen at such low temperate (*e.g.*, < - 30 °C) and render extremely low conductivity and high charge-transfer impedance. While ether-based electrolyte can maintain a decent conductivity, such as the 1 M LiTFSI in DOL/DME system, it poses an extremely increased charge-transfer impedance at subzero temperature (partially due to the large desolvation energy of the dilute ether electrolyte (*ACS Appl. Mater. Interfaces* 2017, 9, 42761–42768; *Joule* 2020, 4, 69–100), which will increase the overpotential when discharging at reduced temperature, leading to poor Li/CF_x performance.

To ensure the clarity of the main figures, we chose to keep the conductivity data of the rest of liquid electrolytes in **Supplementary Fig. 18** with discussions.

Supporting information: page 21

Supplementary Figure 18 | Ionic conductivity of different conventional liquid electrolytes.

Corresponding description has been also added in the main text (Page 12 line 236-241).

Main text: page 12

To further investigate the electrochemical properties of cells employing the MPM with FM, the ionic conductivity was measured by a customized two-electrode conductivity cell (supplemental information). To confirm the reliability of our setups, the ionic conductivity of conventional liquid electrolytes at ultra-low temperatures were conducted for comparison (Supplementary Fig. 18). As presented in Figure 4a, the LGE steadily maintained good conductivity from -60 °C and -30 °C. In contrast, the industry-standard liquid electrolyte (e.g., 1 M LiPF₆ in ethylene carbonate (EC)/diethyl carbonate, 1:1 in volume, 1.2 M LiPF₆ in EC: ethyl methyl carbonate, 3:7 by weight) suffered from rapid conductivity fading with decreasing temperature, suggesting the advantage of using LGE in extremely cold conditions.

Comment #10: “Page line 152: Stability over 3 days is grossly inadequate, the battery will need to survive for years (> 10 years for conventional Li-CFx). Any comment on the long-term effects or on the projected shelf life of the batteries with LGE?”

Response: We are grateful for the reviewer for pointing out this important concern. Owing to its excellent shelf life and negligible self-discharge behavior in conventional liquid electrolytes, CF_x was selected as the model cathode, as demonstrated in the literature such as *J. Am. Chem. Soc.* 2014, 136, 6874–6877; *J. Power Sources*, 2006, 160, 577–584. To evaluate the stability of CF_x electrode in our LGEs system, the capacity of Li//CF_x cells with MPM confined LGEs after storage for different time have been tested, for which no noticeable fade of discharge capacity after storage for two months (**Supplementary Fig. 27**). The negligible capacity fading suggests the electrochemical compatibility and reasonably good shelf-life of our Li//CF_x cells with MPM confined LGEs. Note that the slight variation of capacities between 1, 30 and 60 days storage time might be due to the variations in cell assembly process including mass loadings, electrolyte/electrode thickness variations, gas feeding, and ohmic contact, which are often observed in home-made cells. We agree with the reviewer that shelf life for 10 or more years will be needed for commercial primary cells, which may be achieved with combination of Li//CF_x cells with MPM confined LGEs and standard cell structures such as 18650 cylinders.

Corresponding description has been added in the main text (Page 15 line 306-316).

Main text: page 15

To evaluate the stability of CF_x electrode in the LGEs system, the capacity of Li//CF_x cells with MPM confined LGEs after different storage time were tested, in which no noticeable fade of discharge capacity was found even after storage for two months (Supplementary Fig. 27). The negligible capacity fading suggests the electrochemical compatibility and reasonably good shelf-life of our Li//CF_x cells with MPM confined LGEs. Note that the slight variation of capacities between cells with 1, 30 and 60 days storage time might be due to the variations in cell assembly process, which is often observed in home-made cells. Nevertheless, the above preliminary results together highlight the advantage of MPMs toward confining LGE at reduced pressures for ultra-low temperature applications. Typically, shelf life for 10 or more years will be needed for commercial primary cells, which may be achieved with combination of Li//CF_x cells with MPM confined LGEs and standard cell structures such as 18650 cylinders.

Comment #11: “Line 190. What is the optimum uptake of FM to provide sufficient conductivity? 140 psi is fairly high at room temperature and requires a thick cell casing (still unsafe), which will reduce the specific energy.”

Response: We are grateful for the reviewer for this valuable concern. For the first question, considering the measurement of conductivity at room temperature is challenging and the results are sensitive due to the high internal pressure of FM-contained cells, we have tested the conductivity of MPM in FM at -40°C and different pressures to find the turning point of sudden impedance increase (ionic conductivity decrease). As shown in Fig. R3a, no noticeable change was observed during the stepwise reduction of pressure from vapor pressure to 70 psi and then to 65 psi. However, the impedance increases significantly below 60 psi, which indicates that the

uptake of FM in MOFs is not enough to provide sufficient ion transport at this turning point (60 psi, -40 °C). Therefore, we identified that 70 psi is an optimum pressure to gas update.

As for the second question, FM is selected as our model system to demonstrate the nanoconfinement effect due to its high vapor pressure and excellently electrochemical compatibility. For more practical applications, other gas candidates are still under investigation. Currently, we are also working on dimethyl ether (Me₂O) and diethyl ether (DEE) solvents which has much lower vapor pressure compared with FM (**Fig. R3b**). It can be expected that these novel gas systems could significantly improve the energy density using a lower housing mass. We hope more interesting and systematic results will be generated in our future works.

Comment #12: “Cathode is in contact with MOF but the electrolyte is contained within the pores of MOF. How well good cathode/liquid electrolyte interface be developed?”

Response: We thank the reviewer for the insightful question. We propose that the MOF-confined LGE (quasi-solid electrolyte) share the similar mechanism to solid-state electrolytes. To evaluate the interface between LGE and active species during discharge, the bulk impedance changes of Li//MPM//CF_x cells at -40 °C, 70 psi, and different DoDs have been measured. As presented in Supplementary Figure 24, while charge transfer resistance changes up and down during lithiation of CF_x (due to the formation of insulating LiF and conducting carbon), no noticeable changes can be found in the bulk (Ohmic) impedances, coupled with a stable voltage profile and high capacity as shown in **Figure 5d** in the main text, indicating that a good contact and stable interface were formed between cathode/LGE.

Supplementary Figure 24 | Nyquist impedance of Li//CF_x cell (with 20 wt.% of UiO-66 in the cathode) using MPM at 70 psi, -40 °C and different DoDs. Inset shows the detailed comparison of the impedances at high frequency regions.

Comment #13: “Line 201: What are the values for the diffusion coefficient?”

Response: We thank the reviewer for the suggestion on specifying the diffusion coefficient values of the systems. In **Supplementary Fig. 17**, we have included the translational diffusion coefficient information of bulk FM and MOFs adsorbed FM systems at various temperatures and pressures. In the bulk FM models, the translational diffusion coefficient experienced significant change at a certain pressure in its log-scale value. The transition pressures of bulk FM at 243K, 273K and 298K occurred at the approximated pressure-ranges 190 psi-235 psi ($2.3 \cdot 10^{-3} \text{ cm}^2 \text{ s}^{-1}$ - $6.5 \cdot 10^{-5} \text{ cm}^2 \text{ s}^{-1}$), 295 psi-367 psi ($2.2 \cdot 10^{-3} \text{ cm}^2 \text{ s}^{-1}$ - $1.2 \cdot 10^{-4} \text{ cm}^2 \text{ s}^{-1}$), and 630 psi-705 psi ($1.0 \cdot 10^{-3} \text{ cm}^2 \text{ s}^{-1}$ - $1.5 \cdot 10^{-4} \text{ cm}^2 \text{ s}^{-1}$), respectively. In the adsorbed FM models, it was shown that as the pressure increases, the translational diffusion coefficient of adsorbed FM gradually increases until a certain pressure, after which the diffusion coefficient monotonically decreases with increasing pressure, as detailed in page 11 of the main text. The transition points of curves at 243K, 273K, and 298K occurred at about 10.8 psi ($2.0 \cdot 10^{-5} \text{ cm}^2 \text{ s}^{-1}$), 21.7 psi ($2.3 \cdot 10^{-5} \text{ cm}^2 \text{ s}^{-1}$), and 103 psi-200 psi ($1.9 \cdot 1 \cdot 10^{-5} \text{ cm}^2 \text{ s}^{-1}$ - $2.4 \cdot 10^{-5} \text{ cm}^2 \text{ s}^{-1}$), respectively. It is noted that the transition at 298K is not apparent to see because the FM is approaching its critical properties. Corresponding description have been added in the main text (see page 11, line 216-227)

Main text: page 11

Supplementary Fig. 16a-d show the translational diffusion coefficient of bulk FM and adsorbed FM systems. In adsorbed FM systems, we find that as the pressure increases, the translational diffusion coefficient gradually increases until a certain pressure (the phase transition point), after which the f -factor monotonically decreases with increasing pressure. The reduced intermolecular distances between (gaseous) FM molecules before the transition pressure results in greater attractive forces and an increase in the diffusion coefficient. After the phase transition, the compressed, liquefied FM molecules experiences reduced translational degrees of freedom, and are less diffusive. It is shown that the transition conditions (pressures/translational-diffusion-coefficients) of adsorbed FM models at 243K, 273K, and 298K occurs at about 10.8 psi / $2.0 \cdot 10^{-5} \text{cm}^2 \text{s}^{-1}$, 21.7 psi / $2.3 \cdot 10^{-5} \text{cm}^2 \text{s}^{-1}$, and 103 psi-200 psi / $1.9 \cdot 10^{-5} \text{cm}^2 \text{s}^{-1}$ - $2.4 \cdot 10^{-5} \text{cm}^2 \text{s}^{-1}$, respectively. It is noted that at 25 °C, the decrease in the diffusion coefficient after the phase transition pressure is not apparent, due to the fact that the FM is approaching its critical properties.

Comment #14: “Line 248: The capacity of 940 mAh/g is rather high for a CF_x cathode. Its theoretical capacity is 865 mAh/g.”

Response: We thank the reviewer very much for pointing this out. The value of X in CF_x has been evaluate through XPS, which indicates that the x value in CF_x is 0.979 ± 0.075 (Fig. R4). The theoretical capacity is 857 ± 25 mAh/g based on breaking one C-F bond by supplying one electron. We have double checked and repeated the testing of room-temperature capacity of CF_x electrodes. The capacity-voltage curves were plotted in Fig R5a and b, in which the occasionally observed higher value than the theoretical capacity can be explained by the mechanism proposed by Ahmad and Dubois *et al.* (Carbon 2015, 94, 1061-1070), due to the formation of Li_2F^+ species stabilized by the carbon host. Additionally, the possible capacity from the MOFs, carbon black, or the decomposition of novel LGEs have also been investigated by two control experiments. Firstly, the Li//MPM//MOFs cell with LGEs but without CF_x in the cathode (only MOFs, carbon black, and polymer binder) delivers a less than 2 mAh g^{-1} capacity (Fig. R5c). This reveals that no additional capacity comes from MOFs or carbon black. Secondly, the Li//MPM// CF_x cell with conventional liquid electrolyte (1 M LiPF_6 EC/DEC 1/1 in volume) also delivers a higher capacity (~ 940 mAh/g) than the theoretical value (Fig. R5d), thereby eliminating the possible capacity from the LGE electrolyte decomposition. Therefore, it can be concluded that the occasionally observed slightly higher-than-theoretical capacity should come from CF_x by the consumption of Li^+ to form carbon host stabilized Li_2F^+ species. Considering overall $\sim 9\%$ variation of cell capacities, we have updated the discharge curve with medium discharge capacity as more representative data in the revised manuscript (Figure 5).

Fig. R4. XPS profiles of three parallel CF_x powder samples. The measured x value is 0.979 ± 0.075 in CF_x .

Fig. R5 Discharge characteristics of Li// CF_x cells based on (a) MPM or (b) Celgard 2400 as the separator and LGEs as the electrolyte at room temperature. (c) Li//UiO-66 cells with UiO-66 as the cathode and MPM as the separate at room temperature. (d) Li// CF_x cells with Celgard 2400 as the separator and 1 M $LiPF_6$ EC/DEC 1/1 in volume as the electrolyte at different temperatures.

Reviewer 2

Overall comments: *“This is a very nice paper demonstrating the confinement of a liquid gas electrolyte in a MOF membrane where the MOF membrane has been fabricated into reinforced, continuous materials compositing PVDF with nano sheets of MOF grown directly onto a GO support. LGE is a relatively new concept that one of the authors has demonstrated recently, however the use of nanoconfinement in MOF membranes is novel as far as I can tell. The performance of the LGE based on a fluorinated methane and LiTFSI shows excellent low temperature conductivity even under lower pressure, especially when compared with the same LGE supported on Celgard.”*

Response: We are grateful for the review’s comments on the novelty and level of performance we demonstrated in this work. We have carefully designed and performed new experiment to fully address the specific technical questions detailed below.

Comment #1: *“The battery performance of these mixed membrane electrolytes is very impressive relative to a comparison with the current organic liquid electrolyte which hardly cycles at the low temperatures desired in this work. The comparison of device performance between the 'brick and mortar' MOF/GO membrane and the Celgard shows a slightly better performance but I would have liked to see some discussion of reproducibility here as the difference is not really that significant.”*

Response: We thank the reviewer for this valuable concern. We have double checked and repeated the capacity testing of CF_x electrodes. As shown in (Fig. R5a and b), the cells with MPM as the separator deliver comparable or a little bit high capacity at room temperature (855 ± 60 mAh/g vs 810 ± 70 mAh/g), and both of them show slight variation ($< 9\%$) in cell capacities, which is commonly observed in home-made cells. However, the advantages of the MPM system become more apparent upon discharge at low temperature and low pressure as shown in Figure 5d in the main text.

Fig. R5 Discharge characteristics of Li//CF_x cells based on (a) MPM or (b) Celgard 2400 as the separator and LGEs as the electrolyte at room temperature. (c) Li//UiO-66 cells with UiO-66 as the cathode and MPM as the separate at room temperature. (d) Li//CF_x cells with Celgard 2400 as the separator and 1 M LiPF₆ EC/DEC 1/1 in volume as the electrolyte at different temperatures.

Comment #2: “There is quite a lot of discussion relating to simulations and calculations in the experimental section but I dont see the data from this described in any detail in the manuscript. Perhaps this could be elaborated.”

Response: We thank the reviewer very much for pointing this out. We have added some important details about the procedures of calculations and molecular dynamic simulations into the main text (See page 10 in the main text).

In addition, as presented in page 23 to 25 of the main text, we have described the procedure in quantum calculations (MP2/aug-cc-pVTZ level), and the molecular dynamics modeling, including the modeling procedure [GC minimization, NVT, NPT, Langevin dynamics etc.], the type of thermostats [Nose-Hoover and Langevin dynamics], the simulation timesteps, and the data files information [216 FM molecules with amorphous structure was applied for bulk FM models and the adsorbed-FM/UiO-66 structures were taken from GCMC results]. For Grand Canonical Monte-Carlo (GCMC) simulations, we used 2 million moves to stabilize a system and 1 million moves

were used to calculate the adsorption capacities. Corresponding descriptions have been added in page 23 and 24 of the main text).

Main text Page 10:

Further insights into the microscopic interactions between FM and UiO-66 was acquired from computer simulations (Fig. 3d). Both quantum mechanics (QM) calculations and molecular dynamic simulations were applied. In Supplementary Table 1, we described the intermolecular and intramolecular parameters of UiO-66 and FM, where the FM properties were obtained via QM calculations at the MP2/aug-cc-pVTZ level of theory using the Q-Chem 5.0 electronic structure package.⁵³ Initially, we optimized the UiO-66 starting structure (Supplementary Fig. 14) using molecular dynamics (MD) simulations via the Large-scale Atomic/Molecular Massively Parallel Simulator (LAMMPS) simulation engine.⁵⁴ The loading curves of FM in UiO-66 were then obtained from the optimized structure by means of Grand Canonical Monte-Carlo (GCMC) simulations using the MCCCSTowhee simulation package.⁵⁵ The accuracy of our GCMC approach was confirmed based on comparison of the adsorption isotherms of CH₄ and CO₂ to other published works as shown in Supplementary Figure 15.^{56,57} All simulated adsorption isotherms of FM in UiO-66 at variable temperatures exhibited a classical type I isotherm of micropore adsorption (Fig. 3e), in which UiO-66 achieved a 10% mass uptake of FM at 140 psi and 25 °C, in good agreement with our experiments (9% mass uptake at 140 psi, 25 °C).

Main text Page 23-24:

Computational details. The simulation parameters were described in Supplementary Table 1, where the FM properties were obtained via QM calculations at the MP2/aug-cc-pVTZ level of theory using the Q-Chem 5.0 electronic structure package.⁵³ The UiO-66 structure was initially optimized via MD approach from LAMMPS software⁵⁴ with the starting structure shown in Supplementary Fig. 13, and the procedure was detailed in the following “*MD of FM/UiO-66 and UiO-66 systems*” section. Further, GCMC simulations (the MCCCSTowhee simulation package⁵⁵) were applied to model the molecules’ loading value inside the optimized UiO-66 structure. GCMC is a procedure involving insertion/deletion molecules between a system and a reservoir to eventually make system/reservoir in thermodynamic equilibrium. GCMC theory allows researchers to well-determine the number of absorbed molecules inside an absorbent at defined chemical potential (μ), volume (V) and temperature (T) conditions. In each GCMC computation, 3 million moves were performed, and we tested that convergence was obtained in each simulation. The initial 2 million moves were used to stabilize the system, while the last 1 million moves were used to obtain the relevant statistics and absorption capacities. Besides the FM/UiO-66 models, the adsorption isotherms of CH₄ and CO₂ inside UiO-66 were also done in order to confirm the accuracy of our GCMC approach, as shown in Supplementary Figure14.^{56,57} After adsorbed FM capacities were determined, the system properties were further modeled and calculated via MD simulations, as mentioned in the “*MD of FM/UiO-66 and UiO-66 systems*” section. Details of MD simulations were described as follows.

REVIEWERS' COMMENTS

Reviewer #3 (Remarks to the Author):

This manuscript demonstrated an interesting approach for practical realization of liquefied gas electrolytes for low-temperature batteries. As evidenced by the improved low-temperature ionic conductivity and cell performance, condensing fluoro-methane into a sub-nanometer metal-organic framework has proven an effective way to extend the low-temperature operational capability of liquefied gas electrolytes. The authors have addressed all the comments from the previous reviewers and made substantial revisions to improve the scientific quality of their work. With the efforts, the revised manuscript is well organized, and the statements are well supported by the experimental data. The reviewer would like to recommend publication of the revised manuscript.

Sen Xin